# Altered Mesenchymal Stem Cells Mechanotransduction from Oxidized Collagen: Morphological and Biophysical Observations

**DOI:** 10.3390/ijms24043635

**Published:** 2023-02-11

**Authors:** Regina Komsa-Penkova, Adelina Yordanova, Pencho Tonchev, Stanimir Kyurkchiev, Svetla Todinova, Velichka Strijkova, Mario Iliev, Borislav Dimitrov, George Altankov

**Affiliations:** 1Department of Biochemistry, Medical University Pleven, 5800 Pleven, Bulgaria; 2Tissue Bank BulGen, 1330 Sofia, Bulgaria; 3Department of Surgery, Medical University Pleven, 5800 Pleven, Bulgaria; 4Institute of Biophysics and Biomedical Engineering, Bulgarian Academy of Sciences, 1113 Sofia, Bulgaria; 5Institute of Optical Materials and Technologies, Bulgarian Academy of Sciences, 1113 Sofia, Bulgaria; 6Faculty of Physics, Sofia University, St. Clément Ohnishi, 1164 Sofia, Bulgaria; 7Research Institute, Medical University Pleven, 5800 Pleven, Bulgaria

**Keywords:** mesenchymal stem cells, mechanotransduction, collagen, oxidation, YAP/TAZ, focal adhesion

## Abstract

Extracellular matrix (ECM) provides various mechanical cues that are able to affect the self-renewal and differentiation of mesenchymal stem cells (MSC). Little is known, however, how these cues work in a pathological environment, such as acute oxidative stress. To better understand the behavior of human adipose tissue-derived MSC (ADMSC) in such conditions, we provide morphological and quantitative evidence for significantly altered early steps of mechanotransduction when adhering to oxidized collagen (Col-Oxi). These affect both focal adhesion (FA) formation and YAP/TAZ signaling events. Representative morphological images show that ADMSCs spread better within 2 h of adhesion on native collagen (Col), while they tended to round up on Col-Oxi. It also correlates with the lesser development of the actin cytoskeleton and FA formation, confirmed quantitatively by morphometric analysis using ImageJ. As shown by immunofluorescence analysis, oxidation also affected the ratio of cytosolic-to-nuclear YAP/TAZ activity, concentrating in the nucleus for Col while remaining in the cytosol for Col-Oxi, suggesting abrogated signal transduction. Comparative Atomic Force Microscopy (AFM) studies show that native collagen forms relatively coarse aggregates, much thinner with Col-Oxi, possibly reflecting its altered ability to aggregate. On the other hand, the corresponding Young’s moduli were only slightly changed, so viscoelastic properties cannot explain the observed biological differences. However, the roughness of the protein layer decreased dramatically, from R_RMS_ equal to 27.95 ± 5.1 nm for Col to 5.51 ± 0.8 nm for Col-Oxi (*p* < 0.05), which dictates our conclusion that it is the most altered parameter in oxidation. Thus, it appears to be a predominantly topographic response that affects the mechanotransduction of ADMSCs by oxidized collagen.

## 1. Introduction

The extracellular matrix (ECM) initiates several mechanical cues that are able to activate intracellular signaling events through cell–matrix interactions [1,2,3,4]. It is generally agreed that the quality and quantity of ECM determine its physical parameters, affecting cellular response [5], which also applies to the behavior of stem cells [3,6,7,8,9]. Each organ or tissue provides specific mechanical cues [3] that have to be understood in the context of the entire multicellular organization [5,10]. The situation is different, however, when cells interact with surfaces (in a 2D system), which is often the case with implanted biomaterials. Here the surface roughness and surface energy (unified as nanotopography) play a pivotal role [7,8,9,10]. The surface nanotopography strongly influences osteoblastic proliferation, differentiation, and extracellular matrix protein expression [5]. A line of research proves that surface roughness modification of titanium implants improves bone-to-implant contact [11,12]. Other examples are the nanofibers [13] or other linear structures, where the organization of adhesive sites dictates the cellular response [3]. Collectively, a growing body of evidence suggests that both surface stiffness and surface topography affect cell fate, gene expression, and whole cell cycle progression in various cell types [9,14,15,16]. The cell–matrix interaction is mediated by focal adhesions (FA), the main hub for mechanotransduction, connecting the ECM proteins, integrins, and the cytoskeleton [17]. Focal adhesions, however, develop better on 2D surfaces, driven by the stiffness, topography, and surface energy [18], as well as by the organization of adsorbed adhesive proteins [13,19]. The forces exerted on cell adhesion molecules further regulate the RhoA signaling pathway by controlling the activities of guanine nucleotide exchange factors (GEFs) and GTPase activating proteins (GAPs) [1]. Recently, it has become clear that the intracellular Hippo signaling pathway is the next hub that regulates a number of important biological processes, including cellular proliferation, survival, and differentiation [20,21,22,23], and thus determines organ size and tissue homeostasis [20,21,22]. Originally discovered in Drosophila melanogaster, the Hippo pathway is highly conserved across species, as equivalent genes and their products can be found in mammals [20] as a complex cascade of serine/threonine-protein kinases STK3 and STK4 [2,4]. They form a complex with the adaptor/Salvador protein (SAV1) that can phosphorylate and activate the effector protein, large tumor suppressor 1/2 (LATS1/2). At the same time, it inhibits the transcription cofactors “Yes” associated protein (YAP1) and its transcriptional co-activator with PDZ-binding motif (TAZ) [19,20,24,25,26,27].

A growing body of evidence suggests that YAP/TAZ signaling is the next intracellular key for driving cell behavior via the Hippo pathway. When “off,” the phosphorylated YAP/TAZ retains in the cytoplasm where it could undergo proteolytic degradation [5,19] but when “on,” the unphosphorylated YAP/TAZ moves into the nucleus and binds to transcription factors called TEA DNA-binding proteins (TEAD1–4), regulating various proliferative and pro-survival genes, thus having a critical impact on cell behavior [20,21,22,23,24,25,26].

Research from the past decade has tremendously expanded our knowledge about mesenchymal stem cell (MSC) physiology in response to physical signals in the environment [3,6,7,8]. MSCs are a group of progenitor cells characterized by their ability for self-renewal and directed differentiation [3,28]. Within their local tissue microenvironment or niche, MSCs communicate with the ECM accepting various chemical, physical, and mechanical cues to regulate their fate and behavior [3,7,8,27,29,30,31]. Nowadays, it is generally agreed that MSCs perceive their microenvironment through soluble (diffusible) signals and mechanical cues, such as ECM stiffness, nanotopography, or confined adhesiveness [3,23,24,25,26,27,29]. For example, MSCs have the ability to differentiate into neuroblast, chondrocyte, osteoblast, adipocyte, and numerous other cell types when they reside within matrices that mimic the stiffness of their native substrate [10,20,24,29]. However, it reflects their physiological environment providing specific viscoelastic properties, while the topographic response is less studied—though it is proposed that this may also determine the local response of stem cells toward tissue repair and regeneration [24]. Collagen is the most abundant protein in the ECM, critical for its mechanical properties, including stiffness, roughness, extracellular forces, and topography, thus affecting various cell functions and communications [32,33,34]. Though our knowledge of the composition of natural ECM is continuously growing, the impact of its structural organization on the adjacent cellular microenvironment is not well understood, particularly in pathological conditions [32,35].

Oxidative stress is one such condition known to affect the collagen structure and turnover strongly [36,37], including its extracellular processing [32], and remodeling [38,39]. Although these processes are extensively studied, direct investigations utilizing adsorbed collagen layers are rather sparse. From this point, our recent study showed that the oxidation of adsorbed type I collagen alters its remodeling by stem cells [39], opening the door for further research.

The use of adipose tissue-derived MSCs (ADMSCs) as a cellular model also draws considerable attention as they combine relatively easy availability and less donor site morbidity, and possess the characteristic multi-potency making them very suitable for tissue engineering applications [31,39,40]. Here, we provide morphological and quantitative (morphometric) evidence for the altered mechanotransduction of ADMSCs adhering to oxidized collagen involving both focal adhesions (FA) and YAP/TAZ signaling pathways, aiming to understand better the stem cells’ behavior in conditions of acute oxidative stress.

## 2. Results

### 2.1. Initial Cell Attachment

Glass coverslips were coated with native (Col) or oxidized collagen (Col-Oxi) produced by previously described procedure [41] to follow the initial attachment of ADMSCs after 2 h of incubation in a serum-free medium. This study was focused on the early signaling events (see below) since, at later stages, it was expectable that cells would produce a plethora of matrix proteins that may scramble the collagen effect alone. For the same reason, the serum was omitted from the medium. After incubation, the cells were fixed and permeabilized before being stained according to protocol 1 (see Methods section) for actin (to view the cytoskeleton), vinculin (to visualize the focal adhesions), and the nucleus. Phase contrast pictures of living cells were also captured; representative images are shown in Figure 1. As evident from the low magnification phase contrast pictures (A and D) and the morphometric data in Table 1, ADMSCs attached equally well on both substrates but spread differently: much better on native Col (A) with a spreading area of 216.0 μM^2^ versus 179.6 μM^2^ for the oxidized samples (D). At the same time, the cells display more rounded morphology on Col-Oxi, as pointed at with yellow arrows on (D), and are confirmed quantitatively by the higher CSI index for the oxidized samples (0.33), lowering to (0.25) for the native ones, as presented in Table 1. Note that the CSI index tends to 1.0 for a circle and 0 for a line, thus quantitatively reflecting the tendency for the delayed spreading of ADMSC (more rounded morphology) when attached to Col-Oxi. This effect corroborates with the less actin cytoskeleton development and focal adhesion formation in the oxidized samples compared to native ones, pointed out with white arrows in Figure 1 (B and C, respectively).

The quantitative data for FA formation are presented in Table 2. These confirm again the significantly higher values for ADMSC adhering to native collagen, namely, the number of FA, total FA area, and the mean area per a single FA, amounting, respectively, to 317, 1085, and 3.66 vs. 143, 406, and 2.84 for the oxidized samples. These data were calculated from the images [42,43,44].

We also conducted a study covering the later stages of adhesion. As evident from Figure 2, all these morphological differences did not persist at the 24th hour of incubation, apparent from the phase contrast pictures (A,D), equally developed actin cytoskeleton (B,E), and focal adhesions formation (C,E) on both the Col (upper row) and Col-Oxi samples (bottom row). Quantitative analyses were not performed here. In fact, this result supported our initial desire to focus on the early stages of cellular interaction and signaling events giving the option to evaluate the specific effect of collagen oxidation.

### 2.2. YAP/TAZ Signaling Events

To follow the intracellular signaling events upon adhesion of ADMSCs, we fixed and permeabilized the cells at the second hour of incubation before staining simultaneously for actin, nucleus, and TAZ activity (see protocol 2 in the Section 4.5.3).

Figure 3 shows typical images of cells examined sequentially on the red, blue, and green channels.

As evident from these images (Figure 3) and the supporting quantitative analysis presented in Table 3, the YAP/TAZ activity that coincides with the nucleus is apparently higher for native Col (pointed with arrows in Figure 3B,C) compared to Col-Oxi (pointed with arrows in Figure 3E,F), corresponding to 229.3 versus 178.2 intensity of pixels and ratio 176.2 versus 14.4, respectively, as shown in Table 3. These values suggest a significantly better signal transmission to the nucleus for native collagen samples: the cytosolic TAZ accumulation was only 1.3 pixels versus 12.4 for the Col-Oxi ones. Conversely, the nuclear TAZ accumulation was almost 230 pixels for the native samples vs. 178 pixels for Col-Oxi, resulting in a significantly higher ratio of TAZ nuclei/TAZ cytosol of 176.2 vs. 14.4 for Col-Oxi (*p* < 0.05).

Taken together, this uneven distribution of TAZ between the nucleus and cytosol confirms the markedly different mechanical signal transduction between native and oxidized samples, being significantly suppressed in the latter.

Another interesting observation from these images was the difference in the overall nuclear shape: the nuclei in native collagen were more flattened compared to the Col-Oxi ones (see Figure 3B vs. Figure 3E), suggesting a more significant pressure from the cytoskeleton for ADMSC adhering on native collagen. This observation was partly confirmed by the nuclear size and shape analysis presented in Table 4: the mean nuclear area per cell was significantly lower in native collagen samples compared to the oxidized ones, amounting to 17.0 μM^2^ versus 21.2 μM^2^ (*p* < 0.05). The effect on the nuclear shape was not so pronounced, and we found only a nonsignificant trend for lowering the mean NSI (0.81 versus 0.86) and the change in NAR from 1.68 versus 1.34 for ADMSC adhering to native vs. oxidized collagen, respectively.

### 2.3. Comparative Atomic Force Microscopy (AFM) Study

The morphology and the mechanical properties of adsorbed native and oxidized collagen, compared with the denatured one, were further examined in the nanoscale using AFM.

For these experiments, collagen was adsorbed to glass coverslips at identic conditions with cellular studies (at 37 °C for 1 h). The measurements were performed by AFM operating in contact mode at room temperature in the air.

As shown in Figure 4, the native collagen forms relatively large linear structures (Figure 4A) resembling thick interlaced fibers. More detailed 3D analysis, however, showed that these structures are sooner coarse aggregates growing to the z-direction (Figure 4D,G). These linear structures were much thinner on oxidized collagen samples (Figure 4B), showing a tendency for network formation, combined with less growth in the z-direction (Figure 4E,H). In the denatured collagen samples, most of these structures were absent (Figure 4C,F,I). Thermal denaturation curves (Figure 4J–L) obtained by Differential Scanning Calorimetry (DSC) also confirm the relatively minute structural changes in Col-Oxi, thus matching our previous investigation for the appearance of a small pre-peak at 35 °C, apart from the complete absence of any thermal changes in denatured collagen.

The same AFM scans were used to calculate the roughness values (R_RMS_) and Young’s modules (Ea) to follow the mechanical properties of the obtained collagen features. Force–distance curves taken on the collagen samples were selected manually from the force images. Only indentation curves at the top of the fibrils and the overlap region of the collagen aggregates (Figure 4D–F) were considered. Young’s modulus of the native collagen was 56.6 ± 8 MPa, whereas that for the oxidized collagen was 66.8 ± 5 MPa (Table 5), respectively. However, no significant difference (*p* > 0.05) was evident between the two collagen forms. Conversely, the data in Table 5 shows that upon oxidation, the roughness of adsorbed collagen features drops dramatically (approx. seven times) from 27.95 + 5.1 to 5.51 + 0.8 nm, reflecting a strongly altered ability of the protein to aggregate, apart from the native collagen forming large aggregates with a peak to valley distance of about 28 nm (*p* < 0.05). The denaturation leads to a relatively vague assembly of collagen in aggregates.

## 3. Discussion

It is widely accepted that ECM anchors cells and directs cell functions not only by biochemical signals but also via specific mechanical cues [2,3]. Stem cells receive such cues from their microenvironment in the niche through mechanosensing and mechanotransduction [7,8,10,13,45], where collagens play a significant role [30,32,46]. Collagen is crucial because it determines most of the mechanical properties of the tissues and organs [5,32,40,47,48,49].

It is clear today that the physical cues affect proliferation, self-renewal, and the differentiation of MSCs into specific cell fates [3,8,9]; however, little is known about how these cues work in pathological environments, such as the acute oxidative stress that affects numerous homeostatic parameters in the body [36,37]. Recently, we developed a useful in vitro model to study the effect of collagen oxidation on MSC behavior. More specifically, we used adsorbed collagen of either native or preoxidized form [39] as a substratum for ADMSCs adhesion to follow their behavior under conditions that mimic acute oxidative stress [41]. Using this model, we found that oxidation leads to significant suppression of extracellular collagen remodeling by ADMSCs due to minute changes in collagen structure, which opens the door for further applications of this model. Here we show that it may relate to altered mechanical signal transduction in the cell interior. A reasonable question arises: how do MSCs sense such altered collagen structure?

Collagen binding is primarily provided by integrins, mainly α1β1 and α2β1 but also α10β1 and α11β1 [33,34], with an affinity for RGD and GFOGER-like sequences in collagen molecules [33,34]. Integrins are a family of major cell surface receptors generally involved in mediating the cellular response to ECM binding [5]. Composed of alpha and beta subunits, integrins form structural and functional linkages between the ECM fibrils and the intracellular cytoskeletal linker proteins [34]. Binding to immobilized collagen promotes integrin activation and clustering in focal adhesions, which are further associated with intracellular actin filaments through the above-mentioned linker proteins [5]. One such protein is vinculin, a cytoskeletal constituent associated with cell–cell and cell–matrix junctions. It is the most used marker for focal adhesions in anchoring F-actin to the membrane [46,50]. Our results show that ADMSCs hardly develop vinculin-containing focal contacts upon attachment to oxidized collagen, apart from the native collagen, where these structures are well pronounced. It correlates with the substantially diminished cell spreading and cell polarization, which were monitored once morphologically (Figure 1) and confirmed by ImageJ morphometry analysis (Table 1, Table 2, Table 3 and Table 4). We show a significantly reduced Cell Spreading Area (from 216 to 179 μM^2^) and CSI tending to 0.33 (i.e., to a more circular shape) compared to 0.25 for native Col (Table 1). It has to be noted, however, that this morphological difference was valid only for the initial stages of cell spreading, as at the 24th hour it was no longer observed, and ADSCs attached and spread equally well on both substrates (Figure 2), actually confirming our previous investigation [39]. We are prone to explain it by the constitutive ability of ADMSCs to produce very soon their own matrix, containing many other adhesive proteins capable of obliterating the initial collagen effect. As noted above, this was the reason we focused the present study on the initial stage of cell adhesion and the related signaling events, just to be sure that ADSCs attach to collagen only.

Another interesting finding was the observed tendency for flattening of ADMSCs nuclei in samples with native collagen, while on oxidized ones, the nuclei were larger and visibly rounding—a trend confirmed quantitatively by morphometry analysis (Table 1). It is well documented that focal adhesions and stress fibers generated on stiff substrate transduce mechanical forces to the nucleus, leading to nuclear flattening [5,46,47]. Thus, we got additional evidence for the successful transmission of the mechanical signal to the cell nuclei but this was working better for ADMSC adhering to native collagen than on oxidized collagen. Presumably, and there is proof in this direction, nuclear deformation increases the nuclear import of signaling molecules by decreasing the mechanical restriction in nuclear pores [16,47]. Obviously, it also happens in our system, as judged by the nuclear accumulation of TAZ activity. In fact, using immunofluorescent visualization of the YAP/TAZ signaling cascade (anti-TAZ antibody), we demonstrated its more substantial accumulation in the nuclear region at the second hour of incubation (Figure 3), again valid mainly for the ADMSC adhering to native collagen; moreover, it was confirmed statistically with morphometry analysis (Table 3), showing that on the oxidized samples, the signal was considerably fainter (*p* < 0.05).

However, the question still remains: why does the adsorbed native collagen provide a better signal to ADMSC during both the initial recognition phase and in the subsequent steps of signal transmission to the nucleus? Even if we accept the version that oxidized collagen is partly denatured upon oxidation, there is no direct proof that such collagen is worse recognized by the cells. On the contrary, there is proof that the unwinding of the collagen molecule releases additional RGD sequences, which improve cellular interaction [49]. 

On the other hand, the analysis of the DSC curves suggests relatively minute changes in the collagen structure on oxidation; the thermogram splits with the appearance of an additional transition, with added melting temperatures of 33.6 °C (Figure 4J) to a native transition characteristic for collagen at 40.1 °C (melting) (Figure 4K), confirming our previous investigation [41]. 

This structural change in oxidized collagen, however, is far from the curve of denatured collagen, where the complete absence of temperature transitions was observed (Figure 4L). Data in the literature regarding the binding of cells to collagen are quite extensive and sometimes contradictory. As an abundant ECM protein, collagen binds with at least five different groups of cell receptors, including first integrins but also DDR, Glycoprotein VI, Osteoclast-associated receptor (OSCAR), LAIR-1, and uPARAP/Endo180 [48]; therefore, it is difficult to assume that oxidized collagen is not recognized by cells, which directs our thinking rather to the physical parameters of adsorbed collagen. It was most likely to turn our attention to its viscoelastic properties, which are known to affect mechanotransduction significantly [2,3,4,5,10]. To our surprise, however, the changes in Young’s modules (Ea) for the oxidized collagen were relatively slight, with a deviation in Ea of about 20% in the direction of hardening (Table 2); moreover, these were statistically insignificant (*p* > 0.05).

On deeper analysis, however, we decided that such a fact should not puzzle us, considering that this is an adsorbed protein and the role of the underlying substrate stiffness can hardly be ignored. In contrast, the AFM data demonstrated a significant change in surface roughness measured over the adsorbed collagen molecules: from relatively thick linear structures, characterized as coarse aggregates in 3D images, they visibly switch to much thinner linear features on oxidized samples (Figure 4A–F). Moreover, the calculated roughness values (R_RMS_) showed that upon oxidation, the roughness of adsorbed collagen features drops dramatically to about 5.5 nm (pick to valley distance), compared to 28 nm for native collagen samples (i.e., approx. seven times less) reflecting a significantly altered ability (*p* < 0.05) of oxidized protein to aggregate under these conditions. It has to be noted here that the adsorption of proteins was performed at 37 °C for 1 h, i.e., in conditions identical to the cellular studies, meaning that it represented the natural roughness that cells experience from the substratum. 

Though not directly related to collagen, a line of studies confirms the topographic response of stem cells [7,8,9,11,16,17]. It necessitates the conclusion that the most altered parameter to which cells are exposed in our conditions is the roughness of adsorbed protein, i.e., per se, it is a kind of response of ADMSCs to substrate topography, which determines the impaired mechanotransduction from oxidized collagen.

## 4. Materials and Methods

### 4.1. Collagen Preparation

Collagen type I was produced from rat tail tendon by acetic acid extraction and salting out with NaCl, as described elsewhere [39,41]. After centrifugation at 4000 rpm at 4 °C, the pellets were redissolved in 0.05 M acetic acid. The excess NaCl was removed by dialysis against 0.05 M acetic acid. All procedures were performed at 4 °C. Thus, a nearly monomolecular composition of collagen solution, in which the collagen content approaches 100% of the total dry mass, was prepared. The collagen concentration in the solutions was measured by optical absorbance at 220–230 nm [41].

### 4.2. Collagen Oxidation Procedure

The collagen solution (2 mg/mL) was incubated in 0.05M acetic acid, pH 4.3, with 50 µM FeCl_2_ and 5 mM H_2_O_2_ for 18 h at room temperature, as previously described [41]. The oxidant solutions were freshly prepared and 10 mM EDTA was used to stop the oxidation reaction, followed by intensive dialysis versus 0.05 M acetic acid to remove the excess oxidants. The oxidized collagen, Col-Oxi, was freshly prepared before the experiments.

### 4.3. Cells

Human ADMSCs of passage 1 were received from Tissue Bank BulGen using healthy volunteers with written consent before liposuction. The cells were maintained in DMEM/F12 medium containing 1% GlutaMAX™, 1% Antibiotic-Antimycotic solution, and 10% Gibco Fetal Bovine Serum (FBS), all purchased from Thermo Fisher Scientific (Waltham, MA, USA). Every two days, the medium was replaced until the cells reached approximately 90% confluency to be used for the experiments up to the 7th passage.

### 4.4. Morphological Study

For the morphological observations, collagen (100 µg/mL) dissolved in 0.05 M acetic acid was used to coat regular glass coverslips (12 × 12 mm, ISOLAB Laborgeräte GmbH, Eschau, Germany) for 60 min at 37 °C, placed in 6-well TC plates (Sensoplate, Greiner Bio-one, Meckenheim, Germany). Then, the cells were seeded at 5 × 10^4^ cells/well density in the final volume of 3 mL serum-free medium before being incubated for 2 h or 24 h. In a later case, 10% FBS was added at the end of the 2nd hour. The initial cell adhesion and overall cell morphology were studied at the 2nd hour and imaged under phase contrast using an inverted microscope, Leica DM 2900, or processed for immunofluorescent analysis in two protocols, as follows:

#### 4.4.1. First Protocol (Cell Spreading and FA Formation) 

After incubations (2 or 24 h) the samples were fixed with 4% paraformaldehyde and permeabilized with 0.5% Triton X-1000 before fluorescence staining. Green fluorescent Alexa fluorTM 444 Phalloidin (Invitrogen, Thermo Fisher Scientific Inc Branchburg, NJ, USA) was used to visualize the actin cytoskeleton, while the cell nuclei were stained by Hoechst 33342 (dilution 1:2000) (Sigma-Aldrich/Merck KGaA, Darmstadt, Germany).

Darmstadt, Germany). Focal adhesions were viewed with Anti-Vinculin Mouse Monoclonal Antibody (Clone: hVIN-1, Thermo Fisher Scientific, Waltham, MA, USA) IgG (1:150) followed by fluorescent Alexa Fluor 555 conjugated goat anti-mouse IgG (minimal x-reactivity) antibody (both provided by Sigma-Aldrich) used in dilution 1:100;

#### 4.4.2. Second Protocol (YAP/TAZ Signaling Events)

To follow the YAP/TAZ signaling events, separate samples from the same series were stained with a rabbit polyclonal anti-TAZ antibody followed by green fluorescent Alexa FluorTM 444 conjugated goat anti-rabbit antibody (both provided by Sigma-Aldrich, Merck KGaA, Darmstadt, Germany) used in dilution 1:100, further counterstained for cell nuclei with Hoechst 33342 and red fluorescent Rhodamine Phalloidin (Sigma-Aldrich, Merck KGaA, Darmstadt, Germany) to view actin cytoskeleton, using dilutions as above.

Finally, all samples were mounted upside down on glass slides with Mowiol and viewed for 1st protocol using the blue (nuclei), green (actin cytoskeleton), and red (vinculin) channels of an inverted fluorescent microscope (Olympus BX53, Upright Microscope Olympus Corporation, Shinjuku Ku, Tokyo, Japan)) with objectives UPlan FLN (40×/0.50). TAZ samples were viewed separately in the green (TAZ activity), blue (nuclei), and red (actin) channels. A minimum of three representative images were obtained for each sample. The respective image processing software merged the different colors. All experiments were quadruplicated.

### 4.5. Image Analysis

#### 4.5.1. Quantitative Analysis of Raw Format Images by ImageJ

All image analysis was performed per cell using ImageJ, which provides a wide range of processing and analysis approaches. The fluorescence intensity of the fibrillary arrays was measured based on raw format images of cells captured from at least three separate images under the same conditions. Pixel-based treatments are performed to highlight the regions of interest (ROIs) and allow the removal of artifacts. A default black-and-white threshold was used in the segmentation module. Images of equal size (W:1600 px/H: 1200 px) were examined. All measurements were performed at the respective channel of the two or three colored images.

#### 4.5.2. Quantification of Overall Morphological Parameters

Four metrics were acquired: Spread Area (SA), Cell Shape Index (SCI), Aspect ratio (AR), and Focal Adhesion size. The individual cellular domains were determined by generating binary masks using Otsu’s intensity-based thresholding method from fluorescent actin images. Cellular masks were then used to calculate ADMSC SA and CSI. The CSI was calculated using the formula:CSI = 4π×A/P2
where A is the mean cell area and P is the mean cell perimeter.

With this metric, a line and a circle have CSI values of 0 (indicating an elongated polygon) and 1 (indicating a circle), respectively. AR was calculated as the ratio of the largest and smallest side of a bounding rectangle encompassing the cell. The same counter function was used to calculate the nuclear surface area (NSA) and overall nuclear shape index (NSI) as important morphometric characterization for each cell.

#### 4.5.3. YAP/TAZ Signaling

To quantify the YAP/TAZ nuclear-to-cytosolic ratio, binary masks of the nuclei were generated using Otsu’s intensity-based thresholding method from fluorescent Hoechst images and were superimposed with corresponding actin masks to generate masks that encompass the cytosol yet exclude the nucleus. Fluorescent TAZ images were then superimposed either with the nuclear- or cytosol-only masks to isolate the TAZ signal in the nucleus or cytosol, respectively. Integral TAZ signal intensity was determined in these domains, and their ratio was normalized to the corresponding areas. The ratio of TAZ activity in the nucleus versus cytoplasm was further calculated and compared for both native and oxidized samples.

#### 4.5.4. Quantification of Focal Adhesions (FA)

Focal adhesions were estimated following the procedure described by Horzum et al. The steps of image processing were carried out using ImageJ. Briefly, the raw fluorescent images were processed in several steps, as follows [42]: choose the sliding paraboloid option with the rolling ball radius set to 50 pixels [43]; enhance the local contrast of the image using the following values, block size = 19, histogram bins = 256, maximum slope = 6, no mask, and fast [44]; apply mathematical exponential (exp) to minimize the background further; adjust brightness and contrast automatically; run log3d (Laplacian of Gaussian or Mexican Hat) filter, here we define the size of log3Step 1b of log3d filter as sigma X = 5 and sigma Y = 5); run log3d (Laplacian of Gaussian or Mexican Hat) filter; execute analysis of particles command using the following parameters, size = 50, infinity and circularity = 0.00–0.99.

### 4.6. AFM Studies

AFM imaging and force–distance curves of native, oxidized, and denatured collagen were performed using Atomic Force Microscopy (MFP-3D, Asylum Research, Oxford Instruments, Santa Barbara, CA 93117, USA). All measurements were taken in the air and at room temperature. Silicon AFM tips (Nanosensors, type qp-Bio) of 50 kHz resonance frequency and 0.3 N/m nominal spring constant were used.

For all imaging experiments, collagen solutions were deposited on a clean glass coverslip and incubated at 37 °C for one hour to ensure maximal adsorption for each sample. Afterward, the collagen-coated glasses were washed gently with distilled water to avoid buffer crystallization on the surface. Morphometrical (roughness value) and nanomechanical characterization were accomplished using IgorPro 6.37 software. The mechanical properties of the three types of collagen were assumed by Young’s modulus defined by the force–distance (f–d) curves. The value of the elastic modulus was obtained by fitting the force–indentation data to the Hertz model with the embedded IgorPro software, considering the Poisson’s ratio to be ≈0.5:E = 3F(1 − ν^2^)/4√(rδ^3^)(1)
where F is the applied force on the sample, δ is the indentation depth, r is the tip radius, and E and ν are Young’s modulus and Poisson’s ratio, respectively.

### 4.7. DSC Measurements

DSC measurements were performed using DASM4’s (Privalov, BioPribor, Moscow, Russia) built-in, high-sensitivity calorimeter with a cell volume of 0.47 mL. The collagen concentration was adjusted to 2 mg/mL in 0.05 M acetic acid. A constant pressure of 2 atm was applied to the cells to prevent any degassing of the solution. The samples were heated with a scanning rate of 1.0 °C/min from 20 °C to 65 °C and preceded by a baseline run with buffer-filled cells. Each collagen solution was reheated after cooling from the first scan to evaluate the reversibility of the thermally induced transitions. The calorimetric curve corresponding to the second (reheating) scan was used as an instrumental baseline and was subtracted from the first scans, as collagen thermal denaturation is irreversible. The calorimetric data were analyzed using the Origin Pro 2018 software package.

### 4.8. Statistical Analysis

All experiments were conducted with at least 3 independent series with 3–4 cells per group. One-analysis of variance (ANOVA) followed by Tukey-HSD posthoc tests were performed on all data sets. Error is reported in bar graphs as the standard error of the mean unless otherwise noted. Significance was indicated by *, corresponding to *p* < 0.05.

## Figures and Tables

**Figure 1 ijms-24-03635-f001:**
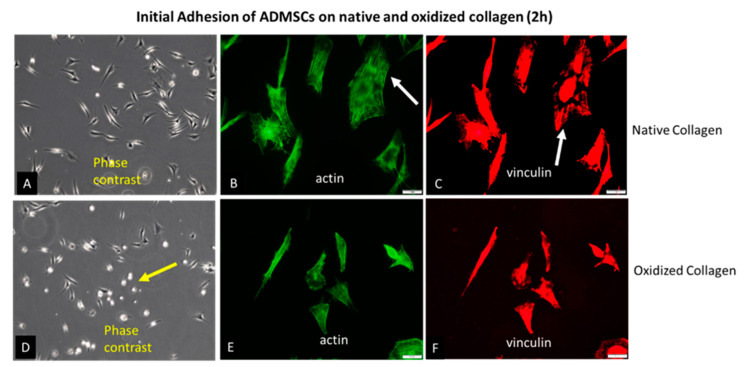
Initial adhesion of ADMSCs to native (**A**–**C**) and oxidized collagen (**D**–**F**) for 2 h in a serum-free medium. The samples were viewed at 10× phase contrast (**A**,**D**) or 40× using the green channel of a fluorescent microscope to view the actin cytoskeleton (**B**,**E**) or red for focal adhesions (**C**,**F**). The yellow arrow on (**D**) points to a delayed spreading of ADMSC on oxidized collagen (more rounded cells), while the arrows on (**B**,**C**) point to the better actin cytoskeleton development and focal adhesions formation in native collagen samples when compared to oxidized counterparts ((**E**) and (**F**), respectively). Bars on (**B**,**C**,**E**,**F**) are 20 μm.

**Figure 2 ijms-24-03635-f002:**
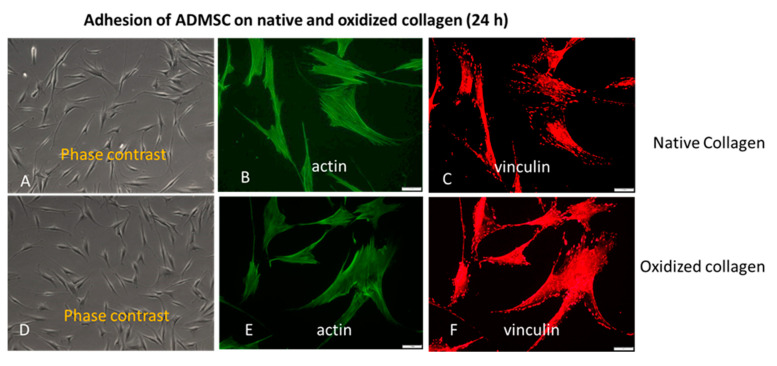
The approximately equal cell spreading (**A**,**D**) corroborates with the similar actin cytoskeleton development (**B**,**E**) and focal adhesions formation (**C**,**F**) at the 24th h of ADMSC adhesion to native (upper row) and oxidized collagen (bottom row). Bar 20 μm.

**Figure 3 ijms-24-03635-f003:**
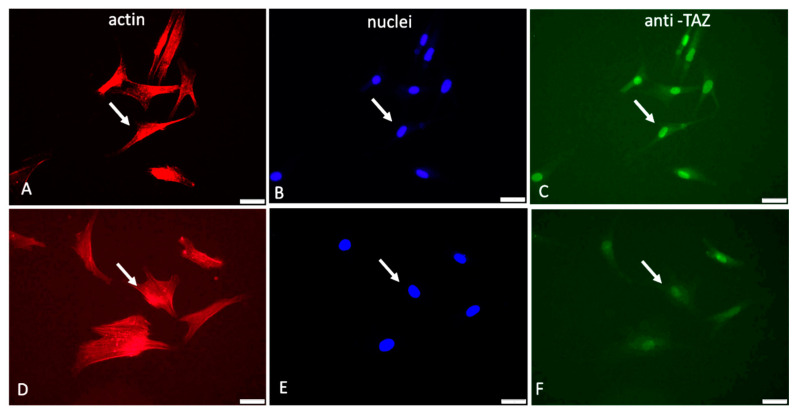
Immunofluorescent visualization of YAP/TAZ activity in ADMSCs adhering to native (**A**–**C**) and oxidized collagen samples (**D**–**F**). The same field was viewed at different channels of the microscope: red (actin cytoskeleton), blue (nucleus), and green (for TAZ activity). Arrows on all the images show the location of the nucleus. Bar 20 μm.

**Figure 4 ijms-24-03635-f004:**
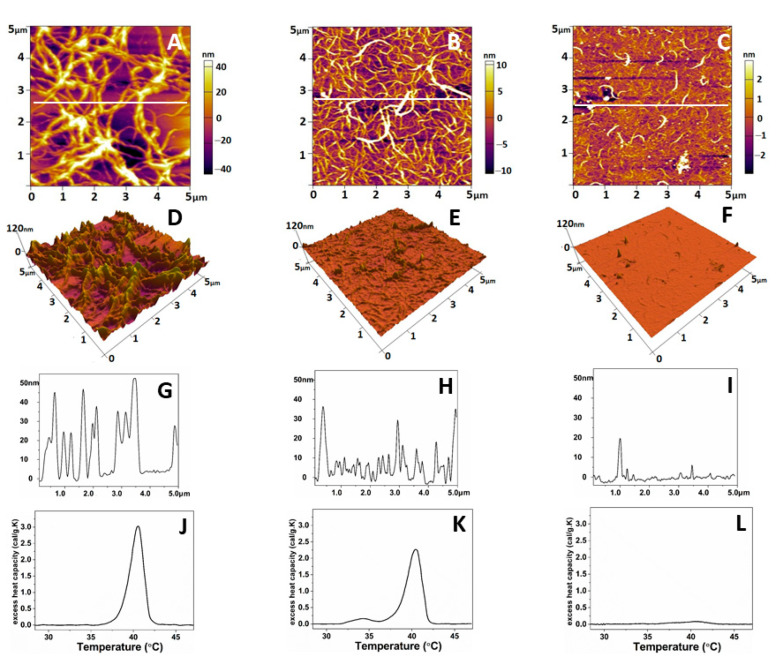
Representative 2D AFM images of native (**A**), oxidized (**B**), and denatured (**C**) collagen; the corresponding 3D topographical images (**D**–**F**) of the images of (**A**–**C**) and cross-section plot shapes (**G**–**I**) corresponding to the white lines in (**A**–**C**). The images were taken in tapping mode in the air at room temperature. The denaturation DSC profiles of native, oxidized, and denatured collagens were presented in panels (**J**–**L**), respectively.

**Table 1 ijms-24-03635-t001:** Morphometric analysis of ADMSC adhering to native and oxidized collagen, including Cell Spreading Area, Cell Shape Index (CSI), Aspect Ratio (AR), and corresponding *p* values.

Cellular Parameters	Col	Col-Oxi	*p*
Cell Spreading Area (μm^2^)	216.0	179.6	*p* > 0.05
Cell Shape Index (CSI)	0.25	0.33	*p* < 0.05
Cell Aspect Ratio (CAR)	3.33	2.61	*p* > 0.05

**Table 2 ijms-24-03635-t002:** Focal adhesion formation: number of FA, total area of FA, and mean area per FA.

Cell	Col	Col-Oxi	*p*
Number of FA	317	143	*p* < 0.05
Total FA Area (μm^2^)	1085	406	*p* < 0.05
Mean Area per FA (μm^2^)	3.66	2.84	*p* > 0.05

**Table 3 ijms-24-03635-t003:** Nuclear and Cytosolic TAZ values and their ratio.

Parameters	Col	Col-Oxi	*p*
Nuclear TAZ	229.3	178.2	*p* < 0.05
Cytosolic TAZ	1.3	12.4	*p* < 0.05
Ratio TAZ Nuclei/TAZ Cytosol	176.2	14.4	*p* < 0.05

**Table 4 ijms-24-03635-t004:** Morphometric analysis of ADMSC nuclei adhering to native and oxidized collagen: nuclear area per cell, nuclear shape index (NSI), Nuclear Aspect Ratio (NAR), and corresponding *p* values.

Nuclear Parameters	Col	Col-Oxi	*p*
Nuclear Area per cell (μM^2^)	17.01	21.2	*p* < 0.05
Nuclear Shape Index (NSI)	0.81	0.86	*p* > 0.05
Nuclear Aspect Ratio (NAR)	1.68	1.34	*p* > 0.05

**Table 5 ijms-24-03635-t005:** Roughness values and Young’s modulus for native, oxidized, and denatured collagen.

Samples	R_RMS_ (nm)	Ea (MPa)
Collagen Native	27.95 ± 5.1	56.6 ± 8
Col-Oxidized	5.51 ± 0.8	66.8 ± 5
Col Denatured	1.04 ± 0.6	3610 ± 59

## Data Availability

Not applicable.

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
