# Peer review of "Altered Mesenchymal Stem Cells Mechanotransduction from Oxidized Collagen: Morphological and Biophysical Observations"

_ijms, 2023, doi:10.3390/ijms24043635_

Round 1

Reviewer 1 Report

The manuscript entitled Altered mesenchymal stem cells mechanotransduction from oxidized collagen. Morphological and biophysical observations was thoroughly reviewed. Adipose tissue derived MSCs (ADМSCs) was taken as cellular model to investigate the altered mechanotransduction of ADMSCs grown under normal or oxidized collagen conditions based on focal adhesions (FA) and YAP/TAZ signaling pathways evidences. It was simple, however, its results were impressive. There are following points to be explained.

1.     In 2.3 Cells of Materials and Methods section, the results would be more sound if 0h and 24h cells were added, it would present a entire process of changes in cells grown under normal and oxidized conditions.

2.     In Discussion section, please add the possible reasons that resulted in altered mesenchymal stem cells mechanotransduction from oxidized collagen.

3.     Some grammars and spelling should be revised.

Page 1 line 25, what is AFM ?

Page 2 line 46, 7-10 should be [7-10].

  Page 3 line 122, substrata?

Page 4 line 140,  withe?

Page 6 line 210, Nuclear Shape Index (NCI) –>NSI

Page 9 line 368, 370Cà370C

Page 9 line 369, 104 cells/wellà 104 cells/well

Author Response

Dear Reviewer 1,

We would like to thank you for your constructive comments. Accepting most of them, we believe it has resulted in an improved version of the manuscript, uploaded alongside the document. We enclose the point-by-point response to your comments for consideration.

Reviewer 1

The manuscript entitled “Altered mesenchymal stem cells mechanotransduction from oxidized collagen. Morphological and biophysical observations” was thoroughly reviewed. Adipose tissue derived MSCs (ADМSCs) was taken as cellular model to investigate the altered mechanotransduction of ADMSCs grown under normal or oxidized collagen conditions based on focal adhesions (FA) and YAP/TAZ signaling pathways evidences. It was simple, however, its results were impressive. There are following points to be explained.

  1. In 2.3 Cells of Materials and Methods section, the results would be more sound if 0h and 24h cells were added, it would present a entire process of changes in cells grown under normal and oxidized conditions.

Comment: This remark sounds reasonable, though only partly. In fact, we thought a lot about the experimental design considering a similar strategy. We did not study "0" hour because there was still no cell adhesion at that time (cells are in suspension), and no signalling events might be expected. Moreover, our morphological approach was not applicable at that time for comparison. On the other hand, a result for the 24th hour would not seem realistic either because, at that time, no morphological difference was observed (Fig. 2), presumably because MSCs (as mesenchymal cells) actively secrete a plethora of matrix proteins that scramble the effect of collagen itself. So, nobody would believe it will reflect the independent effect of collagen oxidation. By the way, we are planning such a study in the context of the question: "how other adhesive proteins would influence this effect of collagen oxidation," but the experimental design is not so simple….

  1. In Discussion section, please add the possible reasons that resulted in altered mesenchymal stem cells mechanotransduction from oxidized collagen.

Comment: We are very grateful to Reviewer 1 for this remark. In the revised version of the Discussion, we added a paragraph (now the fifth) that sheds more light on our understanding of the mechanism. Ultimately, based on the AFM data (showing relatively little change in Young's modulus but substantially reduced roughness) and some literature analysis, we conclude that regardless of the expected response of MSC to elasticity (known to have a substantial effect on mechanotransduction), we ultimately end up with the conclusion for a dominant effect of the roughness in this case, as the most strongly changed parameter upon oxidation.

The latter conclusion also existed in the previous version (last paragraph), but it was obviously not sufficiently explained and linked in the text. Now we believe we have corrected this omission.

  1. Some grammars and spelling should be revised.

Page 1 line 25, what is AFM ?

Comment: We apologize again. The abbreviation Atomic Force Microscopy (AFM) was added to the Abstract and in the text.

Page 2 line 46, 7-10 should be [7-10].

Comment: Corrected.

Page 3 line 122, substrata?

Comment: Corrected to "substrates"

Page 4 line 140, withe? 

Comment: Corrected to "the"

Page 6 line 210, Nuclear Shape Index (NCI) –>NSI

Comment: Corrected to "NSI."

Page 9 line 368, 370Cà370C    

Comment :Corrected to “37 °C”  

Page 9 line 369, 104 cells/wellà 104 cells/well   

Comment: Corrected to 104cells/well

Many thanks to Reviewer I for these very helpful comments.

Reviewer 2 Report

I have read the manuscript with interest and ask for some clarifications;

- please pay attention to some missing or inconsistent abbreviations; for example, the abstract begins with ECM without this abbreviation being defined; page 2 line 46, the square brackets are missing; line 145, Aspect ratio (AP) but in table 1 Cell Aspect ratio (CAR) is indicated; line 210, Cell Aspect ratio (CAR) is reported in table 4 as NAR.

- line 296, your statement "CSI tending to 0.25 (i.e. to more circular shape) compared to 0.33 for Col (Table 1)" should be reversed.

- in the methods, check the degree symbol (37oC) and set the thousands indicator to exponent (104 cells/well density).

- Concerning initial ADMSC adhesion on native vs oxidized collagen, Figure 1 overall shows 14 cells and only one of these shows focal adhesions (Fig. 1C). How many cells in the two conditions showed evident focal adhesions?

- Figures 3D and 3B look blurry to me; please replace them with more focused images.

- Was the illumination intensity of the system the same when the images relating to TAZ activity were taken?

- In 3C and F the cytoplasm signal is almost unseen; how can you differentiate it from background ?

- Morphometric analysis of ADMSC nuclei adhering on native and oxidized collagen does not provide strong evidence of nuclear reshaping. In the discussion the result should be further mitigated (lines 304-307).

Author Response

Dear Reviewer,

We would like to thank you for your constructive comments. Accepting most of them, we believe it has resulted in an improved version of the manuscript, uploaded alongside the document. We enclose the point-by-point response to your comments for consideration.

Reviewer 2

Open Review

Comments and Suggestions for Authors

I have read the manuscript with interest and ask for some clarifications;

- please pay attention to some missing or inconsistent abbreviations; for example, the Abstract begins with ECM without this abbreviation being defined; page 2 line 46, the square brackets are missing; line 145, Aspect ratio (AP) but in table 1 Cell Aspect ratio (CAR) is indicated; line 210, Cell Aspect ratio (CAR) is reported in table 4 as NAR.

Comment: Thank you for these helpful and punctual pieces of advice. All remarks on abbreviations mentioned above were corrected.

- line 296, your statement "CSI tending to 0.25 (i.e. to more circular shape) compared to 0.33 for Col (Table 1)" should be reversed.

Comment: Many thanks for this remark. It is correct, and we apologize for this technical mistake. Now we rewrote the sentence as: "…confirmed quantitatively by the higher CSI index for the oxidized samples (0.33), lowering to (0.25) for the native ones, as presented in Table 1."

- in the methods, check the degree symbol (37oC) and set the thousands indicator to exponent (104 cells/well density).

Comment: It was corrected.

- Concerning initial ADMSC adhesion on native vs oxidized collagen, Figure 1 overall shows 14 cells and only one of these shows focal adhesions (Fig. 1C). How many cells in the two conditions showed evident focal adhesions?

Comment: Because of the still early cell spreading, we have high nonspecific fluorescence in the cells' middle (coming from the push-up of the nucleus). Therefore the focal adhesions (FA) are not very well contrasted. Moreover, the format of all images in this figure is small, for which we apologize. However, still, if you look more carefully, the cells expressing FA in Fig. 1C are 5 (note, with an arrow, we show only a cell with best developed FA), while on 1F (oxidized collagen), we have only one cell, showing rather non-developed vinculin clusters (not arranged as streaks), even not mentioning the smaller size of the cells. Nevertheless, for more convincing (also for us), we performed a comparative morphometric analysis showing a significant difference (p<0.05) for both FA number and total FA аrea.  

- Figures 3D and 3B look blurry to me; please replace them with more focused images.

Comment: Thank you for this remark. We improved the contrast of the entire Figure 3, and we hope the effect of altered TAZ transmission in oxidized samples is more evident now. Nonetheless, since the exposure of the pictures of the oxidized sample (D, E, F) is the same as for native collagen (A, B, C), it precisely emphasizes that the accumulation of TAZ in the nucleus is significantly less, resulting in a more pale image of the nucleus in 3F.

- Was the illumination intensity of the system the same when the images relating to TAZ activity were taken?

Comment: Yes, as mentioned above.

- In 3C and 3F the cytoplasm signal is almost unseen; how can you differentiate it from background ?

Comment: This is true, the cytosolic signal was too low to be visually distinguished, and therefore we performed a quantitative ImageJ analysis.

- Morphometric analysis of ADMSC nuclei adhering on native and oxidized collagen does not provide strong evidence of nuclear reshaping. In the Discussion the result should be further mitigated (lines 304-307).

Comment: We rewrote these paragraphs because we indeed found some inaccuracies in the presentation of the data for morphometry analysis, mainly due to incorrect wording. It was necessary due to the good recommendation of both reviewers.

Finally, many thanks to Reviewer II for the very helpful comments.
